# Who Runs the Most? Positional Demands in a 4-3-3 Formation Among Elite Youth Footballers

**DOI:** 10.3390/s25185825

**Published:** 2025-09-18

**Authors:** Denis Čaušević, Emir Mustafović, Nedim Čović, Ensar Abazović, Cătălin Vasile Savu, Dragoș Ioan Tohănean, Bogdan Alexandru Antohe, Cristina Ioana Alexe

**Affiliations:** 1Faculty of Sport and Physical Education, University of Sarajevo, 71000 Sarajevo, Bosnia and Herzegovina; denis.causevic@fasto.unsa.ba (D.Č.); nedim.covic@fasto.unsa.ba (N.Č.); ensar.abazovic@fasto.unsa.ba (E.A.); 2Football Club Sarajevo, Academy FC Sarajevo, 71000 Sarajevo, Bosnia and Herzegovina; emirmustafovic6@gmail.com; 3Department of Sport Games and Physical Education, “Dunărea de Jos” University of Galați, 800008 Galați, Romania; 4Department of Motric Performance, Transilvania University of Brașov, 500036 Brașov, Romania; 5Department of Physical and Occupational Therapy, “Vasile Alecsandri” University of Bacău, 600115 Bacău, Romania; antohe.bogdan@ub.ro; 6Department of Physical Education and Sports Performance, “Vasile Alecsandri” University of Bacău, 60015 Bacău, Romania; alexe.cristina@ub.ro

**Keywords:** football, stats, GPS, performance, match analysis

## Abstract

This study aimed to examine position-specific physical demands among elite U19 football players competing in a 4-3-3 formation, using data collected via STATSports GPS technology. A total of 23 players from a top-tier Bosnian club, FK “Sarajevo”, were monitored during 26 official matches in the 2024/2025 season. Match data included total distance, distance in six speed zones, high-speed running (HSR), sprint distance, number of sprints, maximum speed, and acceleration/deceleration events. One-way ANOVA and Bonferroni post hoc analyses revealed significant positional differences across all performance metrics (*p* < 0.05). Central midfielders (CMs) covered the greatest total distance and distance per minute, while side defenders (SD) and forwards (FWs) recorded the highest values in sprint distance, HSR, and sprint frequency. Central defenders (CDs) consistently demonstrated the lowest outputs in high-speed and sprint metrics. These findings highlight the distinct physical profiles required for each playing position in a 4-3-3 system and provide practical insights for designing position-specific training and load management strategies in elite youth football.

## 1. Introduction

Modern football is a complex and multifaceted sport, where mental [1,2,3], technical [4,5], tactical [6], and physical [7,8] components interact to shape both the dynamics of the game and its final outcome, stimulating and strengthening the physical and tactical capacities of football players, with particular emphasis on youth development pathways [9], both biologically and in terms of physical literacy, as a component of the development of communication and interrelationship capacities throughout life [10]. As football is a team sport, tactical roles strongly shape the physical demands placed on players, and numerous studies have examined this relationship [4,11,12,13].

Tactics in football are often described as a form of communication—frequently non-verbal—between players, significantly influencing the game’s physical, technical, and cognitive demands [9,14]. Among these, the physical aspect has long been a key focus of sports science research. Football has evolved rapidly in terms of physical demands, particularly regarding high-speed actions [15,16,17]. Numerous studies have investigated players’ fitness profiles [18,19,20,21], differences across age groups [22,23,24,25], and variations by playing position [26,27].

The integration of modern technology has significantly enhanced coaches’ and sports scientists’ ability to monitor performance and understand the physical demands placed on players during matches. Global and local positioning systems (GPSs and LPSs) are widely used to create performance profiles based on running intensities and movement zones [7,8]. This data-driven approach has facilitated improved training planning, load management, and performance optimisation [28,29,30]. Moreover, GPS data supports more nuanced analysis of positional demands [7,8,31], allowing researchers to link tactical behaviour and positional tasks with physical output [32,33,34,35].

Several recent studies have examined performance differences among players within the 4-3-3 formation, including senior professionals [34,36], senior amateurs [37], and youth players [38]. From a performance analysis standpoint, such research allows coaches and scientists to map out physical development pathways aligned with the specific demands of each playing position [39]. This approach also enables the design of training programs and exercises that target external load parameters with precision—whether through tactical drills or isolated training tasks [40].

Despite the growing volume of literature, including systematic reviews and meta-analyses on football performance, there is a notable gap in studies focusing on GPS-derived data from competitive matches involving elite youth players [26,41,42]. Specifically, there is an absence of published research analysing GPS performance in football players from Bosnia and Herzegovina. One major reason for this shortfall is the limited availability and use of GPS technology in both everyday practice and scientific research, despite its recognised value for guiding youth player development. This focus is particularly relevant given that the U19 age group in Bosnia and Herzegovina represents the last stage of academy training prior to integration into senior squads, both at the national and international levels. Moreover, the 4-3-3 tactical formation is one of the most frequently applied systems across European football, including in Bosnia and Herzegovina, and is commonly used in both academy and professional settings [34,36,37,38]. Examining running demands within this formation, therefore, provides not only insights into domestic player development but also allows for meaningful comparison with international research and practical implications for the transition of youth players into senior football.

Previous research has demonstrated clear positional differences in locomotor demands across both youth and professional football. Midfielders consistently cover the greatest total distances due to their continuous involvement in offensive and defensive transitions, while wide defenders and attackers typically perform the highest number of sprints and high-speed runs [43,44,45]. Such evidence informed our hypothesis that side players and midfielders would exhibit the greatest demands in both total distance and high-speed running. In GPS-based performance analysis, key external load variables include total distance, distance covered in predefined speed zones, high-speed running (≥19.8 km/h), sprint distance (≥25.2 km/h), and counts of accelerations and decelerations (>2 m/s^2^). These metrics are widely recognised as sensitive indicators of positional and tactical demands [42,46]. Including them in the present study provides both scientific and applied value for understanding youth football performance.

Therefore, the primary aim of this study is to establish a comprehensive running performance profile of elite U19 football players competing in a 4-3-3 formation and to examine positional differences in physical demands during competitive match play. Drawing from the existing literature—largely based on professional-level players—we hypothesise that (a) side players (wingers and full-backs) and midfielders will cover the greatest total distance and high-speed running distances. Although the hypotheses were formulated on total distance and high-speed running, which represent the most consistently reported variables in football science and allow for direct comparison with the existing literature, the present study also analysed a wider set of performance indicators (e.g., sprint count, accelerations, and decelerations). This ensured a comprehensive assessment of positional demands beyond the primary focus variables.

## 2. Materials and Methods

### 2.1. Experimental Design

This was an observational retrospective study that included footballers from an under-19 (U19) Bosnia and Herzegovina professional team (FK Sarajevo) during 26 matches of the 2024/2025 regular season. The data was obtained through global positioning system (GPS) devices, APEX (STATSports, Newry, UK). The study protocol was approved by the Ethics Committee of the Faculty of Sports and Physical Education, University of Sarajevo, and was conducted in accordance with the ethical principles of the Declaration of Helsinki. Prior to participation, all subjects were fully informed about the aims and procedures of the study and voluntarily agreed to take part.

### 2.2. Participants

Twenty-three professional male football players (18 ± 1 years; height: 183.9 ± 6.2 cm; body mass: 74.3 ± 5.5 kg), competing in the Premier League of Bosnia and Herzegovina and participants in the UEFA Youth League 2024/2025, were included in this study. All the players participated in five training sessions and one match per week. For the main analysis, data from 26 official league matches employing a 4-3-3 formation were used. Matches were excluded if (a) the formation changed during the match, (b) a player was sent off with a red card, or (c) adverse weather conditions (e.g., heavy rain, heavy snow) potentially affected performance. In total, 143 player–match observations were analysed, with players participating in an average of 6.2 matches each (range: 4–8). The distribution of match data by position was central defenders (*n* = 32), side defenders (*n* = 40), central midfielders (*n* = 38), side midfielders (*n* = 17), and forwards (*n* = 16). Only data from players who played for more than 85 min in the same position were included. Goalkeepers were excluded, and playing positions were categorised as central defenders (CDs), side defenders (SDs), central midfielders (CMs), side midfielders (SMs), and forwards (FWs).

### 2.3. Anthropometric and Global Positioning System Variables

Before the start of the study, players’ anthropometric characteristics were assessed. Body height was measured using a digital stadiometer (InBody BSM 370; Biospace Co., Ltd., Seoul, Republic of Korea), while body mass and body composition parameters were evaluated using a direct segmental multi-frequency bioelectrical impedance analyzer (InBody 720; Biospace Co., Ltd., Seoul, Republic of Korea) [47]. GPS variables were collected during official games using APEX (STATSports, Newry, UK). The system’s transmitter was positioned in a custom-designed vest, located on the upper back between the shoulder blades. Once activated, it tracked the player’s movements throughout the match. After the game, the transmitter was inserted into a dedicated device provided by the manufacturer, which transferred the recorded data to a computer. The system has demonstrated high reliability and has been utilised in previous research [13,38]. The following parameters were assessed: total distance (TD); total distance per minute (TD/m); distance zone 1 (DZ1; 0.1–7.19 km/h); distance zone 2 (DZ2; 7.2–10.99 km/h); distance zone 3 (DZ3; 11.0–14.39 km/h); distance zone 4 (DZ4; 14.4–19.79 km/h); distance zone 5 (DZ5; 19.8–25.19 km/h); distance zone 6 (DZ6; ≥25.2 km/h); high metabolic load distance (HMLD); high-speed running (HSR); high-speed running per minute (HSR/m); sprint distance; number of sprints; max speed; accelerations (AC; >2 m/s^2^); and decelerations (DC; <−2 m/s^2^). High-speed running (HSR) was defined as the distance covered at velocities ≥ 19.8 km/h (Zones 5 and 6 combined). Sprint distance and sprint count were calculated based on efforts performed at ≥25.2 km/h (zone 6), with a minimum dwell time of 1.0 s above the threshold applied by the software (Apex, 10 Hz version 4.3.8, STATSports; Northern Ireland, UK) to classify sprinting bouts.

All GPS recordings were obtained under high-quality signal conditions, with the device connected to an average of 15 satellites and an HDOP value consistently below 1.0, indicating excellent accuracy. Only data from players who completed ≥85 min in the same position were included. This threshold was chosen to capture near-complete match exposure and reduce variability associated with substitutions or tactical changes. We acknowledge, however, that a 60 min threshold is widely adopted in the literature, and this methodological decision is discussed as a limitation of the present study.

### 2.4. Statistical Analysis

The statistical analysis was performed using SPSS (Version 22 for Windows, Armonk, NY, USA, IBM Corp.). The Shapiro–Wilk test was employed to assess the normality of the data distribution. All analysed variables met the normality criteria, allowing for the application of parametric statistical procedures. One-way analysis of variance (ANOVA) was conducted to examine intergroup differences in GPS-derived variables across playing positions during matches. When significant effects were identified, Bonferroni post hoc multiple comparisons were used to explore pairwise differences between positions. Effect sizes (ESs) were calculated based on Cohen’s criteria [48]. The magnitude of the effect, represented by the coefficient η^2^, was interpreted as follows: 0.01–0.06 = small effect, 0.06–0.14 = moderate effect, and >0.14 = large effect. Statistical significance was set at *p* < 0.05.

## 3. Results

The anthropometric characteristics of football players by playing position are presented in Table 1. Table 2 provides descriptive statistics (mean ± SD) and 95% confidence intervals for match-specific GPS variables across different positions. The outcomes of the one-way ANOVA, along with post hoc Bonferroni multiple comparison results, are summarised in Table 3. Statistical analyses indicated significant differences across all tested variables between playing positions (*p*-values ranging from 0.001 to 0.006). Post hoc comparisons revealed significant pairwise differences, which are detailed in Table 2.

Regarding distance and pace-related variables, significant differences were observed between playing positions. For high metabolic load distance (HML distance), a significant difference was found between central defenders (CDs) and side defenders (SDs), with CDs covering less distance (*p* = 0.029). Total distance showed significant differences, with central midfielders (CMs) covering more ground than both CDs (*p* = 0.001) and SDs (*p* = 0.007), as well as more than side midfielders (SMs) (*p* = 0.009). Similarly, for total distance per minute, CMs covered significantly more than CDs (*p* = 0.004) and SDs (*p* = 0.011).

Significant differences were observed between playing positions across all detailed distance zones (Zones 1–6). In lower-intensity zones (Zones 1 and 2), central defenders (CDs) and side defenders (SDs) covered more distance than CMs, but less than forwards (FWs) (*p* < 0.001). Conversely, in moderate to high-speed zones (Zones 3 and 4), CMs consistently covered more distance than other positions, particularly more than defenders and forwards (*p* < 0.001). In the highest intensity zones (Zones 5 and 6), SDs and FWs covered significantly more distance than CDs and CMs (*p* < 0.001), indicating distinct positional demands in high-speed running.

Significant positional differences were found across all speed and sprint-related variables. CDs consistently recorded lower values in high-speed running (HSR), HSR per minute, sprint distance, and number of sprints compared to SDs and FWs (*p* < 0.001). SDs showed the highest HSR and sprint activity, significantly outperforming CDs and CMs (*p* < 0.001). CMs exhibited lower maximal speed compared to SMs and FWs (*p* < 0.05). Additionally, sprint distance and the number of sprints were significantly greater in SDs and FWs than in CMs, highlighting the elevated high-speed demands of wide and attacking roles.

Significant positional differences were observed in high-intensity accelerations and decelerations (Zones 3 to 6). CMs demonstrated the highest number of high-intensity accelerations, significantly more than SMs (*p* = 0.001), while FWs also outperformed side midfielders (*p* = 0.008). In terms of decelerations, CDs showed significantly fewer efforts than CMs (*p* = 0.001), and SDs recorded more decelerations than SMs (*p* = 0.045). CMs again exceeded SMs in deceleration efforts (*p* = 0.001), underlining their dominant role in rapid changes of movement during match play.

## 4. Discussion

This study provides an in-depth exploration of the physical demands of elite U19 football players from Bosnia and Herzegovina playing in a 4-3-3 tactical formation. The study focused on positional differences in total distance covered, high-speed running, and sprinting metrics. The findings affirm that central midfielders and wide players exhibit the highest physical demands in terms of both total and high-speed running distances. These outcomes are in line with earlier findings from studies conducted in both youth and professional football contexts [41,43,49,50,51,52]. The results reveal additional information, since nearly 70% of the published papers did not include differences in running performance (RP) between playing positions [42].

CMs covered the highest total distance per match (10,138.63 ± 780.82 m), followed by wide players (9560.41 ± 691.98 m), which aligns with previous studies demonstrating that midfielders tend to cover more distance [46,53]. The results of the present study mirror the activity profiles reported by Pettersen and Brenn [41], where CMs in the U17 competition also covered the highest total distances among all positions. Specifically, Bloomfield et al. [53] highlighted that midfielders, due to their tactical positioning, cover significantly more distance during matches. Additionally, CMs also showed higher total distance per minute (108.99 ± 19.23 m), further emphasising their continuous movement demands [44]. This result aligns with the work of Martin Buchheit et al. [43], who reported that CMs consistently cover more distance compared to other positions in youth football, largely due to their involvement in both attack and defence [46,54]. This does not conclude that higher RP influences success in elite-level football [55].

When comparing the physical demands of U19 players to those observed in senior professional football, the present cohort showed similar total distances but lower values for high-speed running and sprinting metrics. This pattern suggests that while elite youth players are already exposed to professional-level volumes of running, they are still developing the capacity for repeated high-intensity efforts [44,54]. However, differences in the intensity and speed of activities were noted, particularly in the high-speed running metrics, suggesting that while the total distance covered is similar, professional players typically engage in more intense running bouts [44]. The same study observed that older players tend to manage higher intensity and high-speed running due to enhanced aerobic capacity and neuromuscular adaptation, traits that still develop in youth players. Therefore, U19 performance approximates professional players in total distance, whilst performance in high-intensity zones highlights the evolving physical capacity of young athletes as they progress in their development.

The data from our study confirm that wide players and FWs are the most engaged in high-speed running and sprinting activities in a 4-3-3 formation. FWs covered the greatest sprint distances (212.79 ± 133.48 m) and performed over 10 sprints per match, a pattern also observed in professional football [46,54]. Similarly, Modric et al. [44] and Sæterbakken et al. [56] demonstrated that SDs and FWs perform more high-intensity running compared to CDs, which reflects the more explosive and anaerobic demands for these positions. Furthermore, FWs executed a substantial amount of high-intensity actions, as observed in previous research, which reported nearly 30% higher intensity in 4-3-3 vs. other playing formations [45]. In contrast, Morgans et al. [57] and Forcher et al. [34] reported that specific playing formation only affects the running performance of the CD.

These similarities and differences with previous research can be explained by both tactical and contextual factors. For instance, the consistent finding that central midfielders cover the most distance is likely linked to their universal tactical role, which involves continuous support in both offensive and defensive phases regardless of formation or competition level. In contrast, discrepancies in sprinting and high-speed running demands between our findings and those of professional cohorts may arise from differences in tactical systems (e.g., 4-3-3 vs. 4-4-2), the developmental stage of players (U19 vs. senior), and contextual influences such as match tempo, opposition quality, or league-specific playing styles. Recognising these factors helps coaches better understand when positional demands are generalizable and when they may be context-dependent.

The elevated sprint demands of SDs and FWs likely reflect their tactical responsibilities within the 4-3-3 formation. Wide players are frequently tasked with stretching the field, providing overlapping or underlapping runs, and initiating pressing actions in advanced zones. These tactical roles inherently require repeated high-speed and sprinting efforts, often followed by rapid decelerations and defensive recovery. In contrast, CMs have higher total distances, and accelerative actions can be explained by their central involvement in transitions, linking defence and attack, and continuously supporting both flanks. These contextual interpretations emphasise the applied significance of our findings for tailoring position-specific training and tactical conditioning.

The high-speed running performance has been shown to be significantly impacted by possession, formation, and tactic. Notably, players in the 4-3-3 formation, especially wide players, showed higher sprint distances and high-speed running counts, which is consistent with findings that attacking formations (e.g., 4-3-3) tend to place greater demands on wide players in terms of sprinting and high-speed activities [58]. This highlights the importance of position-specific conditioning that focuses on repeated sprint ability and high-speed endurance for wide players and forwards [57].

An important aspect explored in this study was the impact of tactical formations on physical demands. Our findings indicate that wide players and midfielders in a 4-3-3 formation experienced higher physical demands, which aligns with previous studies reporting greater loads for these positions when compared with other tactical systems such as the 4-4-2 [45]. This is consistent with Modric et al. [44], who reported that tactical formation plays a key role in shaping the physical output of players, particularly wide and attacking players. In our study, wide players and forwards displayed greater sprinting distances and higher speed outputs, pointing out the tactical demand of the 4-3-3 formation for these positions. Conversely, the authors of [58] found that CDs playing in more defensive formations, such as the 3-5-2, had a significantly higher number of high-intensity sprints compared to those in formations with four defenders. This difference reflects the variability of demands based on tactical roles, with different formations increasing the physical demands on players in different positions.

Our study also highlighted significant differences in accelerations and decelerations, with CMs and FWs performing the highest number of accelerations, a pattern also observed previously [43,47,54,56,59], who noted that midfielders’ frequent transitions between attack and defence require greater involvement in accelerative movements. Secondly, FWs constantly tend to put pressure on the opponent’s defence, simultaneously trying to gain an advantage in the last third of the pitch. In contrast, wide players, particularly wingers and fullbacks, exhibited a higher frequency of high-intensity decelerations, highlighting their role in covering large distances at high speeds and then quickly changing direction, a characteristic of their tactical responsibilities [54]. This finding is particularly important for designing training protocols that enhance players’ ability to accelerate and decelerate at high intensities, which are crucial for effective performance in transitions, pressing, and defensive recovery actions.

This study is not without limitations. The sample was drawn from a single elite U19 team and covered only one competitive season, which restricts the generalizability of the findings. Tactical demands and physical outputs may vary across seasons due to changes in coaching philosophy, player availability, or broader tactical trends. Additionally, the relatively small number of players in certain positions may have influenced positional comparisons and effect sizes. Future research should therefore expand to include multiple teams across different competitive contexts, examine data across several seasons, and apply longitudinal approaches to better capture how positional demands evolve during the transition from youth to senior professional football.

Another limitation concerns the inclusion threshold of 85 min. Although this criterion was chosen to ensure near-complete match exposure, it differs from the more commonly used 60 min threshold in football GPS research. This methodological choice may influence per-minute metrics (e.g., TD/min, HSR/min), and future studies should compare thresholds systematically to assess their impact on reported performance outcomes.

Future research should further investigate how tactical variations—such as different formations and playing styles—affect the physical demands across specific positions. It is also essential to examine how these demands evolve with age, competitive level, and progression to senior football. Moreover, longitudinal studies are needed to provide a deeper understanding of how repeated exposure to position-specific loads influences long-term player development, performance adaptation, and injury risk.

It should be noted that the running demands observed in this study cannot be solely attributed to the tactical system itself. Individual player characteristics—such as physical stature, maximal sprinting ability, or technical strengths—likely influence both the roles assigned within a formation and the resulting physical outputs. For instance, shorter players may be positioned in roles involving fewer aerial duels, while faster players may naturally be deployed in positions with greater sprinting requirements. Thus, the metrics reported here represent an interaction between tactical context and individual player profiles, underscoring the importance of considering both factors in performance analysis and training design.

## 5. Conclusions

This study provides novel insights into the positional physical demands of elite U19 football players within a 4-3-3 tactical formation. The results clearly demonstrate that players experience significantly different physical and metabolic loads depending on their on-field roles. CMs exhibited the highest total and relative distances covered, reflecting their constant involvement in both offensive and defensive transitions. In contrast, SDs and FWs were found to engage in the highest levels of high-speed running and sprinting, underscoring the explosive demands placed on players occupying wide and attacking roles.

The findings have direct implications for the planning of training and load management in youth football. Coaches and sport scientists should adopt a position-specific approach to conditioning, ensuring that training programs are tailored to the unique demands of each role. Such individualised programming is essential not only for enhancing performance but also for reducing injury risk and supporting long-term athletic development.

The findings of this study provide the first positional benchmarks for U19 players competing in a 4-3-3 formation in Bosnia and Herzegovina. These data can help coaches and sport scientists to design position-specific conditioning programs—for example, emphasising repeated sprint ability and high-speed endurance for wide players and forwards, while focusing on continuous movement capacity and accelerative demands for central midfielders. The results may also guide load management strategies aimed at reducing injury risk by tailoring training intensities to positional profiles. Finally, the benchmarks presented here offer a valuable tool for tracking player development and supporting the transition from youth to senior professional football.

Finally, it is important to recognise that these findings reflect not only the demands of the 4-3-3 formation but also the characteristics of the players within the system. Positional running metrics are shaped by the interplay between tactical structure and individual player profiles, which coaches should consider when interpreting data and planning position-specific training.

## Figures and Tables

**Table 1 sensors-25-05825-t001:** Anthropometric characteristics of participants.

Playing Position	Body Mass (kg)		Body Height (cm)		BMI (kg/m^2^)	
	Mean	SD	Mean	SD	Mean	SD
Central defenders (CDs)	79.25	4.84	190.08	2.89	21.92	1.09
Side defenders (SDs)	71.88	2.41	179.84	2.91	22.24	1.05
Central midfielders (CMs)	74.95	6.95	184.44	6.13	21.96	0.82
Side midfielders (SMs)	66.33	0.28	172.68	0.53	22.24	0.58
Forwards (FWs)	74.52	0.39	188.06	0.25	21.07	0.05
All players	74.35	5.53	183.98	6.21	21.94	0.96

SD = standard deviation; BMI = body mass index.

**Table 2 sensors-25-05825-t002:** Descriptive statistics and 95% confidence intervals for match-specific GPS variables by playing position.

	Central Defenders (CDs) (*n* = 32)	Side Defenders (SDs) (*n* = 40)	Central Midfielders (CMs) (*n* = 38)	Side Midfielders (SMs) (*n* = 17)	Forwards (FWs) (*n* = 16)	All Playing Positions (*n* = 143)
	Mean ± SD 95% CI [−,+]	Mean ± SD 95% CI [−,+]	Mean ± SD 95% CI [−,+]	Mean ± SD 95% CI [−,+]	Mean ± SD 95% CI [−,+]	Mean ± SD 95% CI [−,+]
**Distance and pace metrics**						
**HML distance**	1355.57 ± 333.91 [1235.18, 1475.96]	1609.44 ± 412.13 [1477.64, 1741.25]	1493.23 ± 348.99 [1378.52, 1607.94]	1267.73 ± 367.8 [960.25, 1575.21]	1656.04 ± 286.98 [1503.12, 1808.96]	1501.03 ± 375.36 [1436.89, 1565.16]
**Total distance**	9401.73 ± 627.85 [9175.36, 9628.09]	9560.41 ± 691.98 [9339.1, 9781.71]	10138.63 ± 780.82 [9881.98, 10,395.27]	9173.87 ± 1381.68 [8018.76, 10,328.99]	9531.48 ± 502.27 [9263.84, 9799.12]	9659.96 ± 794.68 [9524.17, 9795.74]
**Total** **distance per** **minute**	98.56 ± 6.7 [96.14, 100.97]	100.04 ± 7.1 [97.77, 102.32]	108.99 ± 19.23 [102.67, 115.31]	107.09 ± 12.6 [96.55, 117.63]	100.56 ± 5.27 [97.76, 103.37]	102.71 ± 12.6 [100.56, 104.86]
**Detailed distance zones**						
**Distance Zone 1**	2888.49 ± 205.47 [2814.41, 2962.57]	2897.84 ± 310.81 [2798.44, 2997.24]	2291.06 ± 189.24 [2228.85, 2353.26]	2772.08 ± 369.68 [2463.02, 3081.13]	3309.32 ± 210.98 [3196.9, 3421.74]	2765.16 ± 411.27 [2694.89, 2835.43]
**Distance Zone 2**	3167.07 ± 322.59 [3050.77, 3283.38]	3082.72 ± 227.06 [3010.1, 3155.33]	3794.43 ± 341.27 [3682.26, 3906.61]	3180.51 ± 566 [2707.32, 3653.69]	2826.06 ± 232.8 [2702.01, 2950.11]	3279.88 ± 458.82 [3201.49, 3358.28]
**Distance Zone 3**	1686.85 ± 223.17 [1606.39, 1767.32]	1597.08 ± 368.17 [1479.34, 1714.83]	2202.63 ± 386.79 [2075.5, 2329.76]	1547.69 ± 292.2 [1303.4, 1791.97]	1519.14 ± 167.99 [1429.62, 1608.65]	1777.99 ± 418.65 [1706.45, 1849.52]
**Distance Zone 4**	1156.31 ± 218.09 [1077.68, 1234.94]	1291.93 ± 336.98 [1184.16, 1399.71]	1432.87 ± 236.93 [1354.99, 1510.75]	1104.56 ± 252.73 [893.27, 1315.84]	1169.9 ± 153.47 [1088.12, 1251.67]	1273.75 ± 282.3 [1225.52, 1321.99]
**Distance Zone 5**	393.21 ± 85.03 [362.56, 423.87]	529.74 ± 147.61 [482.53, 576.95]	364.64 ± 95.63 [333.2, 396.07]	447.21 ± 148.34 [323.2, 571.22]	524.33 ± 112.82 [464.21, 584.45]	444.74 ± 136.45 [421.43, 468.06]
**Distance Zone 6**	109.78 ± 53.44 [90.52, 129.05]	161.09 ± 75.96 [136.8, 185.39]	53.00 ± 32.14 [42.44, 63.56]	121.84 ± 39.39 [88.91, 154.77]	182.74 ± 115.65 [121.11, 244.36]	118.43 ± 80.76 [104.63, 132.23]
**Speed and sprint metrics**						
**HSR**	251.5 ± 57.71 [230.69, 272.3]	345.42 ± 102.67 [312.58, 378.25]	212.88 ± 57.39 [194.02, 231.74]	301.45 ± 66.02 [246.26, 356.64]	353.53 ± 103.66 [298.29, 408.77]	283.75 ± 98.28 [266.96, 300.54]
**HSR per** **minute**	5.26 ± 1.22 [4.83, 5.7]	7.24 ± 2.13 [6.56, 7.92]	4.45 ± 1.21 [4.05, 4.85]	6.54 ± 1.43 [5.35, 7.74]	7.46 ± 2.22 [6.28, 8.64]	5.96 ± 2.07 [5.61, 6.31]
**Max speed**	30.35 ± 1.86 [29.68, 31.02]	30.88 ± 1.51 [30.4, 31.37]	28.84 ± 4.41 [27.39, 30.29]	33.04 ± 7.95 [26.4, 39.68]	31.6 ± 2.01 [30.53, 32.66]	30.39 ± 3.47 [29.8, 30.98]
**Sprint distance**	137.2 ± 70.88 [111.65, 162.76]	201.43 ± 94.21 [171.3, 231.56]	71.14 ± 41.94 [57.35, 84.92]	149.16 ± 45.12 [111.44, 186.88]	212.79 ± 133.48 [141.66, 283.92]	147.38 ± 97.26 [130.76, 164]
**Sprints**	6.66 ± 2.86 [5.63, 7.69]	10.68 ± 4.02 [9.39, 11.96]	3.89 ± 2.02 [3.23, 4.56]	8 ± 2.14 [6.21, 9.79]	10.75 ± 5.27 [7.94, 13.56]	7.64 ± 4.4 [6.89, 8.39]
**Acceleration and deceleration metrics**						
**Accelerations** **>2 m/s^2^**	152.47 ± 41.32 [137.57, 167.37]	153.65 ± 42.16 [140.17, 167.13]	177.71 ± 40.43 [164.42, 191]	112 ± 36.12 [81.8, 142.2]	170.44 ± 29.27 [154.84, 186.03]	159.71 ± 42.5 [152.45, 166.97]
**Deceleration** **<−2 m/s^2^**	124.53 ± 41.11 [109.71, 139.35]	145.93 ± 40.17 [133.08, 158.77]	167.18 ± 43.11 [153.01, 181.35]	103.38 ± 28.5 [79.55, 127.2]	146.69 ± 23.53 [134.15, 159.22]	144.4 ± 42.84 [137.08, 151.72]

*n* = number of player–match observations per position, not number of individual players; HML distance—high metabolic load distance; distance Zone 1: 0.1–7.19 km/h (rest, walking)—very low intensity; distance Zone 2: 7.2–10.99 km/h (jogging)—low intensity; distance Zone 3: 11–14.39 km/h (jogging)—moderate intensity; distance Zone 4: 14.4–19.79 km/h (running)—high intensity 1; distance Zone 5: 19.8–25.19 km/h (fast running)—high intensity 2; distance Zone 6: ≥25.2 km/h (sprinting)—very high intensity; HSR per minute—high-speed running distance per minute of play; max speed—the highest velocity (km/h) reached by the player during the match; sprint distance—total distance (in meters) covered at sprinting speed; sprints—number of sprinting efforts performed; accelerations—greater than 2 m/s^2^; and deceleration—greater than −2 m/s^2^.

**Table 3 sensors-25-05825-t003:** Statistical differences in match-specific GPS variables between playing positions.

	ANOVA			POST HOC			
Variable	F	*p* Value	ES	Bonferroni			
**Distance and pace metrics**							
HML distance	3.787	0.006	0.105	CD < SD (*p* = 0.029)			
Total distance	6.117	0.000	0.159	CD < CM (*p* = 0.001); SD < CM (*p* = 0.007); CM > SM (*p* = 0.009)			
Total distance per minute	4.453	0.002	0.121	CD < CM (*p* = 0.004); SD < CM (*p* = 0.011)			
**Detailed distance zones**							
Distance Zone 1	58.148	0.000	0.643	CD > CM (*p* = 0.001); CD < FW (*p* = 0.001); SD > CM (*p* = 0.001); SD < FW (*p* = 0.001); CM < SM (*p* = 0.001);			
Distance Zone 2	39.406	0.000	0.550	CD < CM (*p* = 0.001); CD > FW (*p* = 0.001); SD < CM (*p* = 0.001); SD > FW (*p* = 0.001); CM > SM (*p* = 0.001);			
Distance Zone 3	23.906	0.000	0.426	CD < CM (*p* = 0.001); SD < CM (*p* = 0.001); CM > SM (*p* = 0.001); CM > FW (*p* = 0.001)			
Distance Zone 4	6.678	0.000	0.172	CD < CM (*p* = 0.001); CM > SM (*p* = 0.013); CM > FW (*p* = 0.008)			
Distance Zone 5	13.199	0.000	0.290	CD < SD (*p* = 0.001); CD < FW (*p* = 0.003); SD > CM (*p* = 0.001); CM < SD (*p* = 0.001); CM < FW (*p* = 0.001)			
Distance Zone 6	17.413	0.000	0.351	CD < SD (*p* = 0.012); CD > CM (*p* = 0.004); CD < FW (*p* = 0.004); SD > CM (*p* = 0.001); CM < FW (*p* = 0.001)			
**Speed and sprint metrics**							
HSR	17.789	0.000	0.356	CD < SD (*p* = 0.001); CD < FW (*p* = 0.001); SD > CD (*p* = 0.001); SD > CM (*p* = 0.001); CM < SM (*p* = 0.041);			
HSR per minute	18.231	0.000	0.361	CD < SD (*p* = 0.001); CD < FW (*p* = 0.001); SD > CD (*p* = 0.001); SD > CM (*p* = 0.001); CM < FW (*p* = 0.001);			
Max speed	4.111	0.004	0.113	CM < SM (*p* = 0.012); CM < FW (*p* = 0.047)			
Sprint distance	15.566	0.000	0.326	CD < SD (*p* = 0.010); CD > CM (*p* = 0.001); CD < FW (*p* = 0.023); SD > CM (*p* = 0.001); CM < FW (*p* = 0.001)			
Sprints	23.672	0.000	0.423	CD < SD (*p* = 0.001); CD > CM (*p* = 0.008); CD < FW (*p* = 0.001); SD > CM (*p* = 0.001); CM < SM (*p* = 0.019);			
**Acceleration and deceleration metrics**							
Accelerations > **2 m/s^2^**	5.594	0.000	0.148	CM > SM (*p* = 0.001); SM < FW (*p* = 0.008)			
Deceleration < −**2 m/s^2^**	7.490	0.000	0.188	CD < CM (*p* = 0.001); SD > SM (*p* = 0.045); CM > SM (*p* = 0.001)			

HML distance—high metabolic load distance; distance Zone 1: 0.1–7.19 km/h (rest, walking)—very low intensity; distance Zone 2: 7.2–10.99 km/h (jogging)—low intensity; distance Zone 3: 11–14.39 km/h (jogging)—moderate intensity; distance Zone 4: 14.4–19.79 km/h (running)—high intensity 1; distance Zone 5: 19.8–25.19 km/h (fast running)—high intensity 2; distance Zone 6: ≥25.2 km/h (sprinting)—very high intensity; HSR per minute—high-speed running distance per minute of play; max speed—the highest velocity (km/h) reached by the player during the match; sprint distance—total distance (in meters) covered at sprinting speed; sprints—number of sprinting efforts performed; accelerations—greater than 2 m/s^2^; and deceleration—greater than −2 m/s^2^.

## Data Availability

The data presented in this study are available upon reasonable request from the corresponding author.

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
