# Peer review of "Who Runs the Most? Positional Demands in a 4-3-3 Formation Among Elite Youth Footballers"

_sensors, 2025, doi:10.3390/s25185825_

Round 1
Reviewer 1 Report
Comments and Suggestions for Authors
Review: Who Runs the Most? Positional Demands in a 4-3-3 Formation Among Elite Youth Footballers
I would like to thank the editorial board for the opportunity to review the current manuscript titled: “Who Runs the Most? Positional Demands in a 4-3-3 Formation Among Elite Youth Footballers”
I commend the authors for investigating the positional demands of elite youth footballers within a 4-3-3 formation. This topic is of value, as understanding the physical requirements of different playing positions is critical for coaches aiming to optimise training, conditioning, and player development. The study is particularly important given the scarcity of published data on U19 players in Bosnia, making this work a valuable contribution to both the regional and wider football science literature.
I would like to see the following points being addressed. In response, please refer to the specific comment and indicate the line number in the revised manuscript.
Thank you.
Introduction
Comment 1: The Introduction provides a clear and well-structured overview of the physical demands of football, recent tactical trends, and the application of GPS technology in performance analysis. There are many citations, which demonstrates the authors’ knowledge of the area.
Comment 2: L60: The introduction could benefit from including the rationale for focusing on Bosnian U19 players and in particular a 4-3-3 formation. For example, why is this sample important? Is this formation the most popular for this age group? How it is relevant for academy player development moving into senior professional football?
Comment 3: The hypotheses focus only on total distance and high-speed running. Could the authors clarify why these two variables were selected over others (e.g., accelerations, decelerations, sprint count), which are also important indicators of positional demands?
Methods
Comment 4: L81: please update the typo: study which included footballers from and under 19 replace with study which included footballers from an under 19
Comment 5: Please include the average number of matches per player, as well as the spread of data across positions. This will provide important context for the sample
Comment 6: The speed thresholds used for high-speed running and sprinting need to be clarified. Are these variables defined using the same thresholds as distance zones 5 and 6, respectively, or are they calculated differently (e.g., based on dwell time in each zone)?
Results
Comment 7: L193: Table 2 – While it is good to see the level of detail presented to two decimal places, I recommend rounding the values to whole numbers. This will make the tables more visually appealing and easier for practitioners to interpret and apply. The same comment applies for the values presented in the Discussion.
Comment 8: L193: Table 2 – Is it necessary to title the variable as “Accelerations Z3 to Z6” when only accelerations >2 m·s⁻² were collected? This wording could be confusing for readers, as it suggests a zonal breakdown that was not presented. A simpler label such as “Accelerations” may improve clarity and avoid misinterpretation.
Comment 9: L193: Table 2 – The n value in the tables currently represents the number of data files per position rather than the number of individual players. This could be confusing for readers, as n is often assumed to represent participants. I suggest clarifying this explicitly in the table legend, or alternatively removing n altogether to avoid misinterpretation.
Discussion
Comment 10: The Discussion effectively places the current findings in context with past research and compares them with professional players. However, it could be enhanced by moving beyond simply restating the results and making comparisons, to include more reflection on why these findings may have occurred.
For example, the higher sprint demands for wide players could be linked to tactical responsibilities such as stretching play, overlapping runs, and pressing. Adding this level of interpretation would provide greater depth and applied value for coaches.
Please review the text and update where necessary
Comment 11: L223: The term high-intensity running is introduced here, but earlier in the Methods (L118) the variable was defined as high-speed running and used in tables as HSR. For consistency, please ensure the same terminology is used throughout the manuscript. Currently, “high-intensity running” appears at L78, 223, 225, and 266, which may cause confusion. It may be that high-speed running (as defined in the Methods) is the correct term, and “high-intensity running” should be amended accordingly.
Comment 12: Can you review the use of “Central midfielders”. You have used the term in L151 then introduced the abbreviations “CM” in L155 but then continued to use the term “Central midfielders” in L157 and in multiple occasions thereafter. Please update
Comment 13: L243-245: It is unclear whether you are stating that U19 players’ physical demands are similar to those of other U19 cohorts in the literature, or whether they are comparable to senior professional players. Please clarify. In addition, please include a reference here for this statement
Comment: 14: L255: I suggest adding the formation context here to make the findings more precise. For example: “The data from our study confirm that wide players and forwards are the most engaged in high-speed running and sprinting activities in a 4-3-3 formation.” Including the formation ensures the reader understands that the demands are formation-specific and may differ in other tactical systems.
Comment 15: L262: The manuscript consistently uses the term high intensity earlier, but here it is abbreviated to HI. For consistency and improved readability, I recommend avoiding the abbreviation and using the full term high intensity throughout the manuscript.
Comment 16: L261-265: You make valuable comparisons to previous studies that both support and contrast the present findings. However, the rationale for these similarities and differences is not fully explored. Please expand on why certain results align with previous work and why others differ, which will provide coaches with a clearer understanding of the context behind the results
Comment 17: L275-276: please add a reference for the 4-4-2 study you have mentioned
Comment 18: L286-289: I recommend removing author names from the main text and instead placing the reference at the end of the sentence. This will improve readability and keep the narrative flowing more smoothly.
For example, the sentence could be revised to:
“Our study also highlighted significant differences in accelerations and decelerations, with central midfielders and attacking players performing the highest number of accelerations, a pattern also observed previously (insert reference here), who noted that midfielders’ frequent transitions between attack and defense require greater involvement in accelerative movements.”
Limitations
Comment 19: The study would benefit from including the study’s Limitations. Currently, several factors that may influence the findings are not acknowledged. For example: the sample includes only one U19 team, the dataset covers a single competitive season, and demands may change across seasons due to tactical trends or player availability and the number of players per position is relatively small, which could bias positional comparisons.
I recommend that the authors acknowledge the limitations of the study and suggest directions for future research.
Practical Applications
Comment 20: The study would benefit from a stronger Practical Applications section, particularly since this is the first study to examine U19 players in a 4-3-3 formation in Bosnia. The findings provide valuable benchmarks that coaches can use to inform position-specific training and
END
Author Response
Dear Reviewer 1,
Time is an extremely precious resource. The authors sincerely thank you for the time you have given to the analysis of the manuscript, the observations and the recommendations.
In attachment we send the responses point by point.
Thank you

Reviewer 2 Report
Comments and Suggestions for Authors
General Comment
The manuscript does not appear to bring novel or original contributions to the scientific community. It is evident that the work relies on previously collected data throughout a season, seemingly with the primary aim of utilizing these data. However, a clear rationale regarding the relevance of the study to the broader scientific field is lacking. The analysis is limited to a single tactical scheme, which restricts the ability to make meaningful comparisons with other systems—something the authors attempt to do in the discussion, but which is not methodologically supported.
Moreover, the rationale for the initial hypothesis is not sufficiently developed in the introduction, nor is the importance of the study or its potential contribution to the scientific community clearly articulated. While I acknowledge the authors’ effort in preparing the manuscript, without a strong rationale and a well-defined research question, the work is reduced to reporting data from a small and very specific sample, which limits its comparability and overall scientific value.
Introduction
The introduction section requires further development. The study hypothesizes that side players and midfielders cover the greatest total distance and high-intensity running distances. However, the rationale supporting this hypothesis is not sufficiently outlined. I recommend expanding the introduction by discussing previous research that has examined locomotor differences across playing positions and tactical formations.
Since the analysis focuses on high-intensity metrics, it is important to provide an adequate background regarding commonly used metrics to assess locomotor performance. Please include a brief overview of these metrics and their relevance.
Line 39-40
“stimulating and strengthening capacities and skills in different categories of people [9]”
Given that your sample consists of football players, the introduction should be more specifically oriented toward the targeted population (e.g., youth or senior footballers).
Line 40-43
“Given that football is inherently a team sport involving the cooperation of 11 players, tactical aspects have received increasing scientific attention [4,10]. Many tactically oriented studies explore the relationship between tactical roles, the physical demands placed on players [10,11], and match performance outcomes [4,12].”
Consider condensing this section to emphasize the direct relationship between tactical roles and physical demands.
Line 59-61
The statement “Despite the growing volume of literature, including systematic reviews and meta-analyses on football performance, there is a notable gap in studies focusing on GPS-derived data from competitive matches involving elite youth players”
Please add appropriate citations.
The manuscript refers to the tactical formation as “1-4-3-3” in some parts and “4-3-3” in others. Please ensure consistency throughout the text.
Methodology
GPS Data Quality
The quality of GPS data is strongly influenced by signal conditions during data collection. An increased number of satellites and a low HDOP value are recommended to optimize GPS accuracy. However, the manuscript does not report the number of satellites available or the HDOP values during data acquisition. Please provide this information.
The rationale for selecting the cut-off value of 85 minutes is not sufficiently justified. Given that the study employs volume-related metrics such as TD/min and HSR/min, match exposure time directly influences these outcomes. As such, the methodological choice raises concerns: either the analysis should be restricted to players who completed the full match, or a lower and more widely adopted threshold (e.g., 60 minutes) should be applied. Furthermore, if the authors decide to use 60-minute criterion, this limitation should be explicitly acknowledged and discussed, particularly with regard to its potential impact on the reported metrics.
Line 119-120
Decelerations (DC; > 2 m/s2).
In physics, only the term acceleration exists, which can be either positive or negative. However, for the sake of clarity, the term deceleration has been adopted, representing a negative acceleration. Accordingly, the threshold must carry the same sign, i.e., < –2 m/s²
Discussion and Conclusions
Line 274-277
“Our findings indicate that wide players and midfielders in a 4-3-3 formation experienced higher physical demands compared to players in setups such as 4-4-2 formation”
This is not supported by the study design. Since the analysis did not include a direct comparison between different tactical formations, this conclusion should be revised.
Line 307-309
“The results clearly demonstrate that players experience significantly different physical, metabolic, and neuromuscular loads depending on their on-field roles.”
In this study, did you directly assess neuromuscular parameters? If you did, you should include those measurements on the manuscript. If not, you should revise this affirmation.
The conclusions of the study are rather narrow and may not accurately reflect the broader reality. Beyond the fact that match metrics are highly dependent on the opponent, it is important to acknowledge that correlation and causation are distinct concepts. One must ask: is it truly the tactical system that generates the reported player metrics, or are these metrics instead shaped by the players’ individual profiles and subsequently aligned with the tactical system? For instance, if a player has a shorter stature, is there not a tendency to place them in roles with fewer aerial duels? With the same purpose, if a player possesses a very high maximum speed, is there not a natural tendency to position them in positions that have more space to run?
This point should be taken into account: it is not solely the tactical system itself that determines outcomes, but also the composition and characteristics of the players who operate within that system.
Author Response
Dear Reviewer 2,
Time is an extremely precious resource. The authors sincerely thank you for the time you have given to the analysis of the manuscript, the observations and the recommendations.
In attachment we send the responses point by point.
Thank you

Reviewer 3 Report
Comments and Suggestions for Authors
When undertaking the review of the scientific article Who Runs the Most? Positional Demands in a 4-3-3 Formation Among Elite Youth Footballers, I read and analysed it with curiosity. It should be noted at the outset that the authors demonstrated their knowledge of the subject matter. Throughout the article, they cited 58 scientific publications related to the research problem. Assessing the entire article, it must be said that it was prepared very professionally and I have no comments for the most part. However, below are a few minor comments that the authors may wish to incorporate into the scientific article:
1. Shouldn't it be 1-4-3-3 instead of 4-3-3, similarly to when we talk about the 3-5-2 and 4-4-2 systems?
2. In the material and method section, line 82, the authors use the words ‘under 19’ and ‘(U19)’ twice - it seems that one of these words is sufficient.
3. If possible, would it be interesting to separate the GPS results into home and away matches? And to show the differences between these matches?
4. In the literature, item no. 41 is missing the entire citation, including the pages.
5. Finally, in the future, it would be interesting to show the difference between two or three teams studied, depending on the playing system used: 1-4-4-2, 1-3-5-2 or 1-4-3-3.
Congratulations on your interesting research paper.
Yours sincerely,
Reviewer
Author Response
Dear Reviewer 3,
Time is an extremely precious resource. The authors sincerely thank you for the time you have given to the analysis of the manuscript, the observations and the recommendations.
In attachment we send the responses point by point.
Thank you

Round 2
Reviewer 2 Report
Comments and Suggestions for Authors
The quality of the manuscript has improved following the successful implementation of the reviewers’ suggestions. Although the topic itself is not entirely original (given the numerous studies examining the influence of tactical formation on positional demands), this paper does present novelty as the first investigation conducted with a sample of U19 players from Bosnia and Herzegovina. Ultimately, the decision on the degree of originality for the journal may be more appropriately determined by the editor-in-chief.
The introduction is now better supported by scientific evidence addressing the relationship between tactical formation and positional physical demands, thereby providing a stronger basis for the stated hypothesis.
The methodological section has also been substantially strengthened. The authors now present detailed information on data collection and analysis procedures, which enhances reproducibility while also acknowledging limitations that should inform future research.
Finally, the discussion and conclusion sections have been refined. The authors avoid comparisons not grounded in the methodology and instead provide a clearer interpretation of the findings, emphasizing that match performance metrics cannot be attributed solely to tactical systems but may also be influenced by players’ physical and technical characteristics.
Author Response
Reply to Reviewer 2 on September 15, 2025
Dear Reviewer 2
We acknowledge the reviewer’s observation that the general topic of tactical formation and positional demands has been examined in prior research. However, we would like to emphasize the novelty of our study in several aspects:
- a) Population specificity – To the best of our knowledge, this is the first investigation conducted with elite U19 players from Bosnia and Herzegovina, a region underrepresented in the literature despite its strong football culture. This expands the geographic scope of existing evidence, which is still largely concentrated in Western European academies.
- b) Age category focus – While most previous studies have analyzed professional senior players, our research targets the elite youth (U19) category, which represents a critical developmental stage where physical demands differ considerably from senior football.
- c) Practical application – The findings provide actionable insights for coaches, conditioning specialists, and academies not only in Bosnia and Herzegovina but also across comparable youth football contexts in Europe and beyond, supporting position-specific training and load management strategies.
Taken together, we believe the study contributes to the literature by addressing both a novel population and an important developmental stage, thereby complementing and extending previous work on tactical formations.
The manuscript has not been modified from the last version, not knowing exactly what to change.
If reviewer 2 still wants adaptation or modification, please specify clearly in which part of the manuscript the authors should respond to his observations.
We thank reviewer 2 for the attention and support offered to the authors!
